

# On the generality of symmetry breaking and dissipative freezing in quantum trajectories

**Joseph Tindall[1,2]⋆, Dieter Jaksch[2,3,4] and Carlos Sánchez Muñoz[5]**

**1** Center for Computational Quantum Physics, Flatiron Institute, New York, United States
**2** Clarendon Laboratory, University of Oxford, United Kingdom
**3** The Hamburg Centre for Ultrafast Imaging, Hamburg, Germany
**4** Institut für Laserphysik, Universität Hamburg, Hamburg, Germany
**5** Departamento de Física Teórica de la Materia Condensada and Condensed Matter Physics Center (IFIMAC), Universidad Autónoma de Madrid, 28049 Madrid, Spain

⋆ jtindall@flatironinstitute.org

## Abstract

Recently, several studies involving open quantum systems which possess a strong symmetry have observed that every individual trajectory in the Monte Carlo unravelling of the master equation will dynamically select a specific symmetry sector to 'freeze' into in the long-time limit. This phenomenon has been termed 'dissipative freezing', and in this paper we argue, by presenting several simple mathematical perspectives on the problem, that it is a general consequence of the presence of a strong symmetry in an open system with only a few exceptions. Using a number of example systems we illustrate these arguments, uncovering an explicit relationship between the spectral properties of the Liouvillian in off-diagonal symmetry sectors and the time it takes for freezing to occur. In the limiting case that eigenmodes with purely imaginary eigenvalues are manifest in these sectors, freezing fails to occur. Such modes indicate the preservation of information and coherences between symmetry sectors of the system and can lead to phenomena such as non-stationarity and synchronisation. The absence of freezing at the level of a single quantum trajectory provides a simple, computationally efficient way of identifying these traceless modes.

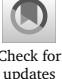

# 1   Introduction

Symmetries are fundamental to our understanding of nature. Quantum physics is no exception to this, and the invariance of the Hamiltonian of a closed quantum system under a given transformation can have far-reaching implications, which include eigenstate degeneracy [1], an absence of thermalisation [2,3] and the protection of topological order [4,5].

In an open quantum system, the influence of an external environment complicates matters and must be accounted for when identifying symmetries and assessing their impact on the system. The last decade has seen significant progress in this regard, with extensive classifications of the symmetries of open systems having been achieved [6–10] alongside an understanding of their physical consequences in terms of steady state degeneracy [11,12], transport properties [13–15], non-stationarity [10,16], quantum synchronisation [17,18], off-diagonal long-range order [19,20] and dissipative phase transitions [21–23]. Results such as this have made it clear that the combination of symmetries and an external environment provides a new pathway for realising unique, exploitable states of matter in a quantum system.

Recently, a new symmetry-based phenomenon, termed 'dissipative freezing', was witnessed in a large quantum spin precessing under the influence of dissipation parallel to the axis of rotation [24]. It was observed how the individual quantum trajectories — solutions to the stochastic Monte Carlo unravelling of the master equation — randomly selected a specific symmetry sector to converge into in the long-time limit; breaking that symmetry at the trajectory level. Due to the simplicity of the model at hand, an analytical proof of this unexpected result was provided, although it was unclear whether this phenomenon could be observed in more general setups. Since this result, several independent works have noted the manifestation of this freezing in other open system setups [25,26], but without offering a deeper explanation for its origin.

In this work, we consider the question of whether dissipative freezing emerges as a general feature of open quantum systems with a strong symmetry. We develop several, simple, mathematical and physical perspectives on the problem to reasonably justify that this is the case, as well as uncovering the direct connection between this phenomenon and previous, rigorous, mathematical proofs establishing the ergodicity of open systems when the external environment is subject to continuous monitoring. We follow these insights up with the derivation of a lower-bound for the rate-of-freezing in terms of the spectral properties of the Liouvillian. Finally, we give several illustrative numerical examples from a range of open systems, where dissipative freezing and the insights developed in this paper can be observed.

Importantly, the simple understanding of dissipative freezing provided in this work allows us to identify the exceptional situations in which, despite the presence of a strong symmetry, dissipative freezing is absent. The most prominent of these situations involves the closing of the

spectral gap in the off-diagonal symmetry subspaces and heralds the emergence of inter-sector, traceless long-lived modes in the Liouvillian's spectrum. These modes ensure the preservation of coherences and information between symmetry sectors despite the environmental influence and can lead to complex physical phenomena such as non-stationarity and synchronisation [10, 17, 18]. By framing this gap closing in terms of an absence of dissipative freezing, we provide a novel, computationally efficient method for identifying the presence of such inter-sector traceless modes in open systems.

## 2 Theory

### 2.1 The GSKL Equation and the Strong Symmetry

Our starting point is a quantum system coupled to an environment under the Markov approximation. The most general completely positive trace-preserving (CPTP) map for the evolution of this system's density matrix $\rho(t)$ at time $t$ is given by the Gorini-Sudarshan-Kosakowski-Lindblad (GSKL) equation [27, 28]:

$$\frac{\partial \rho(t)}{\partial t} = \mathcal{L}\rho(t) = -\mathrm{i}[H, \rho(t)] + \sum_{j=1}^{M} \gamma_j \left[ L_j \rho(t) L_j^\dagger - \frac{1}{2}\{L_j^\dagger L_j, \rho(t)\} \right], \tag{1}$$

where $H$ is the system's Hamiltonian and $L_1, L_2, ..., L_M$ are a series of 'jump' operators which describe the interaction of the system with the environment. We set $\hbar = 1$ throughout and use $\gamma_j$ to set the dissipation strength.

Due to the addition of the dissipative term in the equation of motion of the system, symmetries can no longer simply be identified by operators which commute with the Hamiltonian $H$ and there are, in fact, several different types of symmetry that an open system can possess [6]. Here, we focus on the 'strong symmetry', which can be identified by a Hermitian operator $A$ that satisfies

$$[H, A] = [L_j, A] = [L_j^\dagger, A] = 0 \ \forall j, \tag{2}$$

ensuring the observable $\langle A \rangle$ is a constant of motion.

As $A$ is Hermitian, we can diagonalise it

$$A = \sum_{\alpha=1}^{D} \lambda_\alpha \sum_{\beta=1}^{d_\alpha} |\alpha, \beta\rangle \langle \alpha, \beta|, \tag{3}$$

where the index $\alpha$ runs over the $D$ distinct eigenvalues $\lambda_\alpha$ of $A$ and $\beta$ runs over the corresponding $d_\alpha$ degenerate eigenvectors $|\alpha, \beta\rangle$ for a given $\alpha$, i.e. $A|\alpha, \beta\rangle = \lambda_\alpha |\alpha, \beta\rangle$. Throughout this work we will use $\alpha$ to notate a 'subspace' or 'symmetry subspace' spanned by these degenerate vectors, and define a 'block' as the operator space spanned by the $d_\alpha d_{\alpha'}$ elements $|\alpha, \beta\rangle \langle \alpha', \beta'|$ with $\beta = 1...d_\alpha$ and $\beta' = 1...d_{\alpha'}$. We refer to a given block as 'diagonal' if $\alpha = \alpha'$ and 'off-diagonal' otherwise.

Now, as the Hamiltonian and the jump operators all commute with $A$ it immediately follows that $A(O|\alpha, \beta\rangle) = \lambda_\alpha(O|\alpha, \beta\rangle)$ for $O = H, L_j, L_j^\dagger \ \forall j$ and so they cannot map an eigenstate of $A$ out of its subspace. They are thus all 'block-diagonal', meaning we can write them in the following form

$$O = \sum_{\alpha=1}^{D} \sum_{\beta, \beta'=1}^{d_\alpha} O_{\alpha, \beta, \beta'} |\alpha, \beta\rangle \langle \alpha, \beta'|. \tag{4}$$

The presence of a strong symmetry therefore immediately implies there exists a simultaneous block-decomposition of the Hamiltonian and jump operators. We emphasize that the converse

is also true: if there exists some basis $\{|\alpha, \beta\rangle\}$ in which the Hamiltonian and jump operators are simultaneously block-diagonal then there exists a strong symmetry of the system given by the operator $A$ in Eq. (3) — i.e. the operator which is just some scalar multiple of the identity matrix in each diagonal block and zero everywhere else.

Due to this strong symmetry there will be multiple steady states of the system, i.e. multiple density matrices $\rho_\infty$ which satisfy $\mathcal{L}\rho_\infty = 0$ [6]. In fact, the basis of steady states (the nullspace of the Liouvillian) will contain at least $D$ trace unity Hermitian steady states. This is because the projection of the Liouvillian into the superoperator space formed from two copies of a given diagonal block constitutes a valid CPTP map and that via Evans' theorem there will exist at least one trace unity steady state for such a map [29, 30]. There can also exist traceless steady states, or even traceless imaginary eigenmodes (modes $\rho_{\text{imag}}$ with imaginary eigenvalues — i.e. $\mathcal{L}\rho_{\text{imag}} = i\lambda\rho_{\text{imag}}$, $\lambda \in \mathbb{R}_{\neq 0}$) of the Liouvillian. In the case that these traceless modes are the eigenvectors of the projection of the Liouvillian into the superoperator subspace formed from two copies of a given off-diagonal block, we will refer them as 'inter-sector traceless, non-decaying modes'. This is because they are solely comprised of coherences between states in different symmetry sectors, i.e. they have $\text{Tr}(\rho |\alpha, \beta\rangle \langle \alpha', \beta'|) \neq 0$ for $\alpha \neq \alpha'$. Later in this work we will demonstrate how such modes prevent dissipative freezing.

## 2.2 The Monte-Carlo Unravelling of the GSKL Equation and Dissipative Freezing

In order to fully introduce dissipative freezing we must describe the Monte-Carlo unravelling of the GSKL equation in Eq. (1). This is an unravelling of the master equation in Eq. (1) into a stochastic equation of motion at the level of pure states [31–33]. After statistical averaging of the independent solutions to this equation of motion, known as quantum trajectories, the exact dynamics of the density matrix of the system is guaranteed to be recovered. We emphasize that this unravelling of the GSKL equation is not just a mathematical formalism. If the action of the jump operator can be ascribed to a property of the environment which is measurable with a POVM (positive-operator-valued-measure), then it is possible to physically realise the stochastic equation of motion which governs the trajectories by subjecting the environment to continuous measurement[1] [32, 34–36]. As a result, the phenomenon of dissipative freezing that we will describe is relevant from both a mathematical and a physical perspective.

Given an initial pure state of the system, $|\psi(0)\rangle$, all independent solutions to the stochastic equation of motion which unravels Eq. (1) can be written, to first order in the propagation timestep $\Delta t$, in the following form

$$|\psi^{(i)}(t)\rangle = \frac{1}{\mathcal{N}_i} \prod_{k=1}^{n} X_k^{(i)} |\psi(0)\rangle , \tag{5}$$

where $i$ is used to index the given trajectory, $\mathcal{N}_i$ is a normalisation constant, $t = n\Delta t$ is the time with $n \in \mathbb{Z}_0^+$ and

$$X_k^{(i)} \in S = \left\{ \sqrt{\gamma_1 \Delta t} L_1, ..., \sqrt{\gamma_M \Delta t} L_M, 1 - iH_{\text{eff}}\Delta t \right\} = \{S_1, ..., S_{M+1}\}, \tag{6}$$

with $H_{\text{eff}} = H - \frac{i}{2}\sum_j \gamma_j L_j^\dagger L_j$. The state $|\psi^{(i)}(n\Delta t)\rangle$ is proportional to $X_n^{(i)} |\psi^{(i)}((n-1)\Delta t)\rangle$ and the operator $X_n^{(i)}$ which propagated it from time $(n-1)\Delta t$ to time $n\Delta t$ is determined stochastically from the set $S$ with the respective 'draw' probabilities being

$$p_m = \begin{cases} \langle\psi^{(i)}((n-1)\Delta t)|S_m^\dagger S_m|\psi^{(i)}((n-1)\Delta t)\rangle, & m \leq M, \\ 1 - \sum_{m=1}^{M} \langle\psi^{(i)}((n-1)\Delta t)|S_m^\dagger S_m|\psi^{(i)}((n-1)\Delta t)\rangle, & m = M+1. \end{cases} \tag{7}$$

---

[1]For instance, if the jump operator is the photon annihilation operator then photon counting measurements in the environment would constitute the POVM.

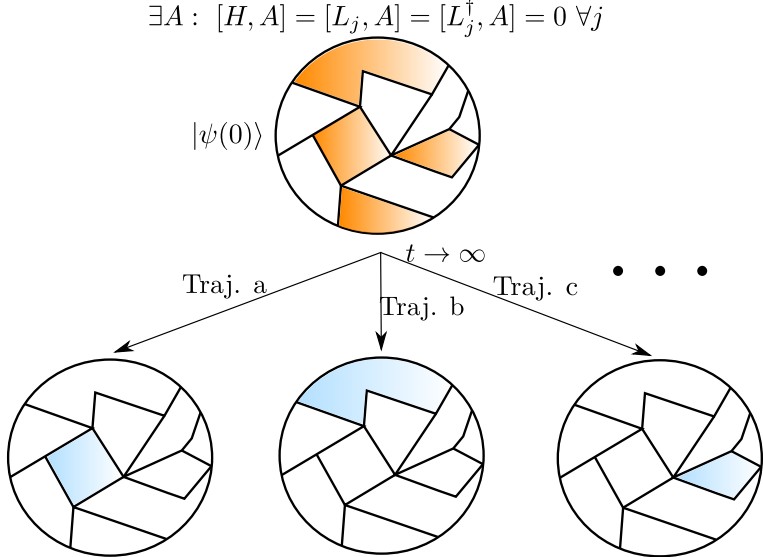

Figure 1: Dissipative Freezing: The Hilbert space of an open quantum system fragments into a series of subspaces in the presence of a strong symmetry operator *A*. For an initial state in a superposition of states in these different subspaces the long-time dynamics of a single quantum trajectory will almost always involve a 'freezing' into one of these subspaces selected at random.

The evolution of a single trajectory is thus a Markovian process: the choice of operator $X_n^{(i)}$ is dependent only on the previous state $|\psi^{(i)}((n-1)\Delta t)\rangle$.

The density matrix of the system at time $t$ — i.e. the solution to Eq. (1) — can then be approximated as a sum over $N$ independent trajectories: $\rho(t) \approx \frac{1}{N}\sum_{i=1}^{N} |\psi^{(i)}(t)\rangle\langle\psi^{(i)}(t)|$ and in the limits $\Delta t \to 0$ and $N \to \infty$ this approximation becomes exact [32]. The unravelling of Eq. (1) we have just described is accurate to first order in $\Delta t$. We opt for this first order unravelling in this paper as it is illustrative and all of our arguments are valid for arbitrarily small $\Delta t$. Unravellings which are of a higher order in $\Delta t$ involve letting the wavefunction evolve freely under $\exp(-iH_{\text{eff}}\Delta t)$ until its norm decays below a certain threshold, at which point it is acted on with a randomly chosen jump operator.

Now that we have introduced the Monte Carlo unravelling of the GSKL equation, we can explicitly describe the phenomenon of dissipative freezing:

> Dissipative Freezing: If an open system governed by the GSKL equation possesses a strong symmetry $A$ then, regardless of the initial state of the system, in the limit $t \to \infty$ a given quantum trajectory will, generically (i.e. outside of just a few known exceptional cases which will be specified in Section 2.5), only have non-zero coefficients in one symmetry subspace.

We should emphasize that the given subspace into which freezing occurs will vary with each trajectory, allowing the system to recover the full density matrix — which could have non-zero coefficients in multiple diagonal blocks — upon averaging. The probability of freezing to a given subspace will therefore be given by the initial norm of the wavefunction in that subspace. Figure 1 provides an illustration of the phenomenon of dissipative freezing.

In the case when the initial state of the system is already confined to one symmetry subspace, then the phenomenon of dissipative freezing is immediate and trivial. Each trajectory will always remain in that subspace as the operators $X_k^{(i)}$ are block-diagonal and cannot map

it out of the given subspace. This also means that once a trajectory is frozen, it will remain frozen for any future times. When the initial state of the system is in a superposition of states in different subspaces, then dissipative freezing is non-trivial. It implies a dynamical breaking of the strong symmetry at the level of trajectories — with the system randomly selecting one of the (possibly) degenerate subspaces of $A$ in the long-time limit. As we will see, such a symmetry breaking is a direct consequence of the presence of a strong symmetry and means that the system preserves the symmetry at the ensemble level by breaking it at the level of pure states.

### 2.3 Insights into the Emergence of Dissipative Freezing

**Recovering a Block-diagonal Steady State**

The simplest case to be made for dissipative freezing can be understood from the form of the steady state. Assuming there are no inter-sector traceless non-decaying eigenmodes of the Liouvillian, then this steady state can always be written in block diagonal form

$$\rho_\infty = \lim_{t \to \infty} \exp(\mathcal{L}t)\rho(0) = \sum_{\alpha=1}^{D} \sum_{\beta,\beta'=1}^{d_\alpha} \rho_{\alpha,\beta,\beta'} |\alpha,\beta\rangle \langle \alpha,\beta'|, \tag{8}$$

just like the jump operators and Hamiltonian. We know that, if $\Delta t \to 0$ in our trajectories unravelling, upon averaging over an infinite number of quantum trajectories in the long-time limit we must exactly recover this steady state, i.e.

$$\rho_\infty = \lim_{t \to \infty} \rho(t) = \lim_{t \to \infty} \lim_{N \to \infty} \frac{1}{N} \sum_{i=1}^{N} |\psi^{(i)}(t)\rangle \langle \psi^{(i)}(t)|, \qquad |\psi^{(i)}(t)\rangle = \sum_{\alpha,\beta} c_{\alpha,\beta}^{(i)}(t) |\alpha,\beta\rangle, \tag{9}$$

where the $c_{\alpha,\beta}^{(i)}(t)$ are the coefficients of the $i$th trajectory at time $t$ for the basis vector $|\alpha,\beta\rangle$. Equations (8) and (9) therefore combine to put a constraint on the coefficients of the trajectories in the long-time limit via

$$\lim_{t \to \infty} \lim_{N \to \infty} \frac{1}{N} \sum_{i=1}^{N} c_{\alpha,\beta}^{(i)}(t) c_{\alpha',\beta'}^{*\,(i)}(t) = 0, \qquad \forall \alpha \neq \alpha', \tag{10}$$

ensuring the steady state is block-diagonal in the long-time limit. It is clear that if dissipative freezing occurs then Eq. (10) is satisfied as there is only one $\alpha$ for which the trajectory coefficients in the long-time limit will be non-zero, implying $c_{\alpha,\beta}^{(i)}(t) c_{\alpha',\beta'}^{*\,(i)}(t) \equiv 0 \,\forall i$ and $\alpha \neq \alpha'$. Without dissipative freezing, Eq. (10) requires the product $c_{\alpha,\beta}^{(i)}(t) c_{\alpha',\beta'}^{*\,(i)}(t)$ to ensemble-average[2] to 0 without the individual terms being 0.

This is very restrictive. The symmetry subspaces are essentially independent open systems with the total Hamiltonian and jump operators being direct sums of the Hamiltonian and jump operators for each subspace. Thus, despite being independent, the subspaces constrain each other via Eq. (10). A given trajectory initialised in a superposition of subspaces must obey this constraint as $t \to \infty$ and also ensure coefficients in the same subspace ensemble average to the appropriate steady state elements, i.e.
$(1/N)\sum_i c_{\alpha,\beta}^{(i)}(t) c_{\alpha,\beta'}^{*\,(i)}(t) = \rho_{\alpha,\beta,\beta'}$ as $t \to \infty$. The more subspaces that are included in the problem (and this number is arbitrary considering they are just independent open systems), the more constraints appear on the trajectories as $t \to \infty$. Dissipative freezing is a natural way

---

[2]We define ensemble-average as the stochastic averaging over all trajectories.

of satisfying this constraint, independently of the number of trajectories used to approximate the steady state, the number of subspaces and the structure of the subspaces.

Moreover, dissipative freezing means we cannot use a single trajectory to simultaneously infer information (however approximate) about the steady state properties of multiple subspaces. Given their complete informational independence (if the steady state is block-diagonal it contains no information — coherences — involving both subspaces simultaneously) this makes sense. We must instead run multiple trajectories, with each providing information about a separate subspace, in order to infer such information — just as if we were to simulate these independent subspaces separately.

**Long-Products of Matrices**

For our second insight, and an illustration of how dissipative freezing emerges as a dynamical process, we consider the explicit time evolution of a single trajectory. To help with our analysis, we introduce the projection operator $P_\alpha = \sum_{\beta=1}^{d_\alpha} |\alpha, \beta\rangle \langle \alpha, \beta|$ which projects into the subspace spanned by the eigenvectors of the strong symmetry $A$ with eigenvalue $\lambda_\alpha$. For a given trajectory at time $t$ we can form the projection into the $\alpha$ subspace as $|\psi_\alpha^{(i)}(t)\rangle = P_\alpha |\psi^{(i)}(t)\rangle$. Due to the block-diagonal nature of the Hamiltonian and jump operators, it follows from Eq. (5) that

$$|\psi_\alpha^{(i)}(t)\rangle = \frac{1}{\mathcal{N}_i} \prod_{k=1}^n X_{\alpha,k}^{(i)} |\psi_\alpha(0)\rangle, \tag{11}$$

where $X_{\alpha,k}^{(i)} = P_\alpha X_k^{(i)} P_\alpha$. The norm of the wavefunction in Eq. (11) can then be interpreted as the probability $p^{(i)}(\alpha, t)$ of the trajectory $i$ being observed in the subspace $\alpha$ at time $t$. We scale this by the normalisation constant $\mathcal{N}_i$ (which is the same for each subspace and so irrelevant to our analysis) to define the un-normalised weight

$$w^{(i)}(\alpha, t) = \langle \psi_\alpha(0) | (A_\alpha^{(i)}(t))^\dagger A_\alpha^{(i)}(t) | \psi_\alpha(0) \rangle, \qquad A_\alpha^{(i)}(t) = \prod_{k=1}^n X_{\alpha,k}^{(i)}. \tag{12}$$

In the limit $t \to \infty$ Eq. (12) is the expectation value of an infinitely long product of matrices. Long products of matrices have been studied extensively in mathematics and physics due to their relevance to dynamical systems and chaos [37–39]. When building up repeated matrix products such as $A_\alpha^{(i)}(t) = X_{\alpha,n}^{(i)} X_{\alpha,n-1}^{(i)} X_{\alpha,n-2}^{(i)} \ldots X_{\alpha,1}^{(i)}$, the quantities of interest are usually the singular values $\sigma_{\alpha,q}^{(i)}(n) := \sigma_q(A_\alpha^{(i)}(t))$ of the product with $q$ indexing them from largest to smallest. These singular values are intrinsically related to the powers of the matrix norm $||A_\alpha^{(i)}(t)||^q$ whose logarithm is often known as the $q$th Lyaponuv exponent. In general, such quantities will grow or decay exponentially with $n$ [40, 41], causing the largest singular value $\sigma_{\alpha,1}^{(i)}(n)$ to become dominant when taking expectation values and allowing us to write

$$(A_\alpha^{(i)}(t))^\dagger A_\alpha^{(i)}(t) \approx \left(\sigma_{\alpha,1}^{(i)}(n)\right)^2 P(\sigma_{\alpha,1}^{(i)}(n)), \tag{13}$$

where $P(\sigma_{\alpha,1}^{(i)}(n))$ is a projector to the subspace (not to be confused with the subspaces indexed by $\alpha$) spanned by the (possibly degenerate) eigenvectors of $(A_\alpha^{(i)}(t))^\dagger A_\alpha^{(i)}(t)$ with the associated eigenvalue $\left(\sigma_{\alpha,1}^{(i)}(n)\right)^2$. Provided the initial state in a given pair of symmetry subspaces $\alpha_1$ and $\alpha_2$ has a non-zero weight in their respective highest singular-value subspaces then the ratio of the probabilities $p^{(i)}(\alpha_1, t)$ and $p^{(i)}(\alpha_2, t)$ will be proportional to the ratios of the corresponding singular values, i.e.

$$\frac{w^{(i)}(\alpha_1, t)}{w^{(i)}(\alpha_2, t)} = \frac{p^{(i)}(\alpha_1, t)}{p^{(i)}(\alpha_2, t)} \propto \left(\frac{\sigma_{\alpha_1,1}^{(i)}(n)}{\sigma_{\alpha_2,1}^{(i)}(n)}\right)^2. \tag{14}$$

If these singular values each vary exponentially with $n$, however, then the ratio of probabilities will clearly either vanish or diverge in the limit $t \to \infty$ depending on whether the growth rate of the leading singular value is largest in $\alpha_1$ or $\alpha_2$. One symmetry subspace thus becomes completely dominant and the wavefunction effectively vanishes in the other subspaces. Freezing thus occurs between all symmetry subspaces with distinct growth rates as a natural consequence of the exponentially growing nature of $A_\alpha^{(i)}(t)$.

We should emphasize that our statement of an exponential growth in the leading singular values $\sigma_{\alpha,q}^{(i)}(n)$ is intended to be that of an average change by orders of magnitude in linear time, i.e. the quantity $\frac{2}{n} \ln(\sigma_{\alpha,q}^{(i)}(n))$ (which can be interpreted as the average growth rate of the $q$th singular value) does not asymptotically tend to 0 for large $n$. We have been careful not to assign a specific growth rate to the singular values $\forall n$ and instead talk about the 'average' change as we are working with matrices and only under certain conditions can one assign a single growth rate $\forall n$ to the leading singular value of long products [37]. Nonetheless, the average change of orders of magnitude of the singular values of $(A_\alpha^{(i)}(t))^\dagger A_\alpha^{(i)}(t)$ in linear time will, via Eq. (14), be reflected in the $p^{(i)}(\alpha, t)$ and cause them to differ by increasing orders of magnitude as time increases linearly. In our numerical results we will observe this growth explicitly.

**Ergodicity and Quantum Systems Subject to Repeated Measurement**

For a final perspective behind dissipative freezing, we will discuss its connection between established mathematical results on the ergodicity of open systems. Specifically, work in the early 2000s successfully proved the ergodicity of an open quantum system undergoing continuous measurement in the environment when there is a single unique steady state [42, 43]. This ergodicity guarantees that the time-average of observables associated with a single trajectory between times 0 and $\tau$, with $\tau \to \infty$, can be used to recover the expectation values associated with the steady state.

In this work it was also realised that, if the steady state of the system is dependent on the initial state in some manner, then this ergodic theorem cannot be proven [42]. The presence of a strong symmetry in an open system guarantees the existence of multiple steady states and thus a dependence of the steady state on the initial state $\rho(0)$ via its weights in the different subspaces. It therefore follows from Ref. [42] that an ergodic theorem cannot be established in the presence of a strong symmetry — i.e. the time-averaged dynamics of a single trajectory cannot be used to recover the ensemble average over multiple trajectories.

This lack of a standard ergodic theorem for a system with degenerate steady states implies that something else must be at play in order for the ensemble average over trajectories to recover the dynamics of the open system. This was uncovered in Ref. [44], where Kuemmerer and Maassen proved that, in this degenerate case, a single trajectory will establish ergodicity solely with respect to one of the randomly selected steady states. By observing that the existence of a strong symmetry in an open system implies multiple steady states and a series of symmetry subspaces, the work of Ref. [44] provides an ergodic understanding and proof of the phenomenon of dissipative freezing. By freezing into one of the symmetry subspaces, the trajectory of an open system can establish ergodicity and — upon time averaging — recover the steady state solely within that subspace.

Moreover, given an initial state $|\psi(0)\rangle$ the probability of freezing into a specific subspace will be proportional to its weight in that subspace, i.e. $p_\alpha = \langle\psi(0)|P_\alpha|\psi(0)\rangle$. This ensures the ensemble average over trajectories will recover the steady state $\rho_\infty$ appropriate to the initial state, i.e. one where $\mathrm{Tr}(P_\alpha \rho_\infty) = \langle\psi(0)|P_\alpha|\psi(0)\rangle$ in each subspace and $\langle A \rangle$ is conserved between times 0 and $\infty$.

## 2.4 The Timescale of Dissipative Freezing

With an understanding of the generality of dissipative freezing, we now tackle the question of the timescales on which dissipative freezing will occur. To aid us in our analysis, we work in the channel-state duality picture, converting operators into supervectors via

$$O = \sum_{\alpha=1}^{D} \sum_{\beta,\beta'=1}^{d_\alpha} O_{\alpha,\beta,\beta'} |\alpha,\beta\rangle \langle \alpha,\beta'| \to ||O\rangle\rangle = \sum_{\alpha=1}^{D} \sum_{\beta,\beta'=1}^{d_\alpha} O_{\alpha,\beta,\beta'} |\alpha,\beta\rangle |\alpha,\beta'\rangle, \quad (15)$$

and constructing the the Liouvillian superoperator

$$\hat{\mathcal{L}} = -\mathrm{i}(H \otimes 1 - 1 \otimes H) + \sum_j \gamma_j (L_j \otimes L_j^* - \frac{1}{2}(L_j^\dagger L_j \otimes 1 + 1 \otimes L_j^T L_j^*)), \quad (16)$$

which acts on these supervectors. As all of the operators involved in the construction of the Liouvillian are simultaneously block-diagonal, the Liouvillian will be too, meaning it can be expressed in the form

$$\hat{\mathcal{L}} = \sum_{\alpha,\alpha'} \sum_{\beta,\beta'\beta'',\beta'''} L_{\alpha,\alpha',\beta,\beta',\beta'',\beta'''} |\alpha,\beta\rangle |\alpha',\beta'\rangle \langle \alpha,\beta''| \langle \alpha',\beta'''|. \quad (17)$$

There are thus $D^2$ diagonal blocks in the superoperator picture and they can be indexed by the tuples $(\alpha, \alpha')$ [45].

Let us now focus on a specific block $(\alpha_1, \alpha_2)$ and extract the corresponding $d_{\alpha_1} d_{\alpha_2} \times d_{\alpha_1} d_{\alpha_2}$ submatrix of the Liouvillian

$$(\hat{\mathcal{L}})_{\alpha_1,\alpha_2} = \sum_{\beta,\beta'\beta'',\beta'''} L_{\alpha_1,\alpha_2,\beta,\beta',\beta'',\beta'''} |\alpha_1,\beta\rangle |\alpha_2,\beta'\rangle \langle \alpha_1,\beta''| \langle \alpha_2,\beta'''|. \quad (18)$$

We define the following spectral gap

$$\Delta^{(\alpha_1,\alpha_2)} = -\sup(\{\mathrm{Re}(\lambda_1), \mathrm{Re}(\lambda_2), ..., \mathrm{Re}(\lambda_{d_{\alpha_1} d_{\alpha_2}})\}) \geq 0, \quad (19)$$

where $\lambda_1, \lambda_2, ..., \lambda_{d_{\alpha_1} d_{\alpha_2}}$ are the eigenvalues of $(\hat{\mathcal{L}})_{\alpha_1,\alpha_2}$. The quantity $\Delta^{(\alpha_1,\alpha_2)}$ essentially dictates the longest timescale of the dynamics of the system in the $(\alpha_1, \alpha_2)$ block. If $\alpha_1 = \alpha_2$ then $\Delta^{(\alpha_1,\alpha_2)}$ is guaranteed to be 0 due to the existence of at least one steady state for that symmetry sector. If $\alpha_1 \neq \alpha_2$ then we refer to $\Delta^{(\alpha_1,\alpha_2)}$ as the inter-sector spectral gap and it is not necessarily zero. In fact, the assumption that the steady state of the open system is block-diagonal is equivalent to the statement $\Delta^{(\alpha_1,\alpha_2)} > 0 \; \forall \alpha_1 \neq \alpha_2$, meaning that all inter-sector coherences will decay away with time.

Now we consider the dynamics of the density matrix of the system in this sector, which is given by $P_{\alpha_1} \rho(t) P_{\alpha_2}$. We will hereon assume $t$ is sufficiently large such that all decay modes other than the longest-lived one have decayed away to give

$$||P_{\alpha_1} \rho(t) P_{\alpha_2}\rangle\rangle = c \exp(-\Delta^{(\alpha_1,\alpha_2)} t) \exp(\lambda t)||\rho_R\rangle\rangle, \quad (20)$$

where $\lambda$ is an arbitrary purely imaginary number, $c = \langle\langle \rho_L ||\rho(0)\rangle\rangle$ and $||\rho_L\rangle\rangle$ and $||\rho_R\rangle\rangle$ are the left and right eigenvectors of $(\hat{\mathcal{L}})_{\alpha_1,\alpha_2}$ corresponding to the longest-lived mode. We can work in the trajectories picture here by substituting Eq. (9), giving us

$$\lim_{N\to\infty} \frac{1}{N} \sum_{i=1}^{N} c_{\alpha_1,\beta}^{(i)}(t) c_{\alpha_2,\beta'}^{(i)}(t) = c \exp(-\Delta^{(\alpha_1,\alpha_2)} t) \exp(\lambda t) \rho_R^{\alpha_1,\beta,\alpha_2,\beta'}, \quad (21)$$

with $\rho_R^{\alpha_1,\beta,\alpha_2,\beta'} = \langle \alpha_1,\beta| \langle \alpha_2,\beta'|||\rho_R\rangle\rangle$.

Taking the norm of both sides we can invoke the triangle inequality to arrive at

$$|c|\exp(-\Delta^{(\alpha_1,\alpha_2)}t)|\rho_R^{\alpha_1,\beta,\alpha_2,\beta'}| \leq \lim_{N\to\infty}\frac{1}{N}\sum_{i=1}^{N}|c_{\alpha_1,\beta}^{(i)}(t)c_{\alpha_2,\beta'}^{(i)}(t)|. \tag{22}$$

Now we define the quantity $\tilde{p}^{(i)}(\alpha,t) = \sum_\beta |c_{\alpha,\beta}^{(i)}(t)|$ — which is larger than the standard probability $p^{(i)}(\alpha,t)$ — sum over $\beta$ and $\beta'$ and use the fact $\sum_{\beta,\beta'}|\rho_R^{\alpha_1,\beta,\alpha_2,\beta'}| > 1$ as the right eigenvectors are normalised to arrive at

$$|c|\exp(-\Delta^{(\alpha_1,\alpha_2)}t) \leq \lim_{N\to\infty}\frac{1}{N}\sum_{i=1}^{N}\tilde{p}^{(i)}(\alpha_1,t)\tilde{p}^{(i)}(\alpha_2,t) = \overline{\tilde{p}(\alpha_1,t)\tilde{p}(\alpha_2,t)}, \tag{23}$$

where the overbar represents statistical averaging.

The product $\overline{\tilde{p}(\alpha_1,t)\tilde{p}(\alpha_2,t)}$ which emerges here is in fact a very natural quantity with which to analyse dissipative freezing as, for $t \to \infty$, it vanishes in its presence and remains finite in its absence. Equation (23) dictates a bound on the rate at which it can decay and tells us that, assuming all but the longest-lived eigenmode(s) has decayed away, it cannot fall below a certain value until a certain time. For our numerics in the next section we therefore define the 'freeze-time' as the earliest time $t_f$ where $\overline{\tilde{p}(\alpha_1,t)\tilde{p}(\alpha_2,t)} < \epsilon$ and $\epsilon$ is a finite, small number, $\epsilon \ll 1$. It is the time at which we consider freezing to have occurred between the two subspaces $\alpha_1$ and $\alpha_2$. Following Eq. (23), for $\epsilon$ sufficiently small and $N$ sufficiently large, this definition freeze-time is lower-bounded as

$$t_f > \frac{1}{\Delta^{(\alpha_1,\alpha_2)}}(\ln(|c|) - \ln(\epsilon)), \tag{24}$$

providing us with an approximate timescale for observations of freezing.

Along with the bound we are able to derive on it, this definition of the freeze-time is reasonable because for a single trajectory initialised solely within the subspaces $\alpha_1$ and $\alpha_2$ we have that, on average, one of $p^{(i)}(\alpha_1,t_f)$ or $p^{(i)}(\alpha_2,t_f)$ will be on the order of $\epsilon^2$ and thus neglibible if $\epsilon$ is chosen to be sufficiently small. Such a result can be seen by observing $\left(\tilde{p}^{(i)}(\alpha,t)\right)^2 > p^{(i)}(\alpha,t)$ and $p^{(i)}(\alpha_1,t) + p^{(i)}(\alpha_2,t) = 1$ at all times and that, at the freeze-time, $\tilde{p}^{(i)}(\alpha_1,t_f)\tilde{p}^{(i)}(\alpha_2,t_f) = \mathcal{O}(\epsilon)$.

We remark that it might be more desirable to achieve an inequality involving $\overline{p(\alpha_1,t)p(\alpha_2,t)}$ than $\overline{\tilde{p}(\alpha_1,t)\tilde{p}(\alpha_2,t)}$. Without making stricter assumptions about the Liouvillian, however, we are unable to achieve this and thus focus on $\overline{\tilde{p}(\alpha_1,t)\tilde{p}(\alpha_2,t)}$ when analysing the 'freezing time' in our numerics.

## 2.5 Exceptions to Dissipative Freezing

There are 'exceptional' cases where the conservation of $\langle A \rangle$ and the recovery of the appropriate steady state is achieved within the trajectories formalism without dissipative freezing occurring. Here we identify the two cases we are aware of and which follow naturally from the mathematical arguments of Section 2.3.

**Similar Symmetry Subspaces**

Consider two subspaces $\alpha_1$ and $\alpha_2$ which are of the same dimension and where the following is true

$$U(H)_{\alpha_1}U^\dagger = (H)_{\alpha_2}, \qquad U(L_j)_{\alpha_1}U^\dagger = \exp(i\theta_j)(L_j)_{\alpha_2}, \quad \forall j \quad \theta_j \in \mathbb{R}, \tag{25}$$

with $U$ being an arbitrary $d_{\alpha_1} \times d_{\alpha_2}$ unitary matrix and $(O)_{\alpha_i}$ indicating that we have extracted the $d_{\alpha_i} \times d_{\alpha_i}$ submatrix from $O$ that corresponds to the $\alpha_i$ diagonal block. Eq. (25) essentially

states that the relevant matrices (Hamiltonian and jump operators) for the two subspaces are identical up to an arbitrary change of basis and a phase factor on the jump operators. If this is true, then freezing cannot occur between subspaces $\alpha_1$ and $\alpha_2$. This is straightforwardly proven by observing that $\left((A_{\alpha_1}^{(i)}(t))^\dagger A_{\alpha_1}^{(i)}(t)\right)_{\alpha_1} = \left((A_{\alpha_2}^{(i)}(t))^\dagger A_{\alpha_2}^{(i)}(t)\right)_{\alpha_2}$ at all times and thus the weights in the two subspaces cannot diverge with respect to one another. We term the symmetry subspaces 'similar' in this case and emphasize that the steady state is still block-diagonal in this case, i.e. $\Delta^{(\alpha_1,\alpha_2)} \neq 0 \ \forall \alpha_1 \neq \alpha_2$ and $\overline{\tilde{p}(\alpha_1,t)\tilde{p}(\alpha_2,t)}$ will decay to 0 in the long-time limit as per Eq.(23) and Eq. (10) is fulfilled without dissipative freezing.

We emphasize that, although we are currently unaware of any, other forms of similarity than that expressed in Eq. (25) may exist between pairs of subspaces which prevents freezing.

**Inter-sector Traceless Modes**

The second, less trivial, case occurs when there are 'inter-sector' traceless non-decaying eigenstates of the Liouvillian, i.e.

$$\exists \alpha_1, \alpha_2, \qquad \alpha_1 \neq \alpha_2, \qquad \text{s.t. } \Delta^{(\alpha_1,\alpha_2)} = 0. \tag{26}$$

By 'inter-sector' we mean that such states possess coherences $|\alpha_1,\beta\rangle \langle \alpha_2,\beta'|$ between states in different symmetry subspaces and do not decay away in the long-time limit. These non-block-diagonal modes imply, via Eq. (23), $\overline{\tilde{p}(\alpha_1,t)\tilde{p}(\alpha_2,t)}$ cannot decay to 0 for an appropriate choice of initial state and therefore freezing cannot occur. We also note that Eq. (26) is equivalent to stating that Eq. (10) is not valid. In general, determining whether Eq. (26) holds requires diagonalisation of the Liouvillian and is computationally expensive – although we remark that Eq. (26) is guaranteed to hold if one can identify an operator, known as the strong dynamical symmetry [10,45], which satisfies $[H,S] = -\lambda S$ and $[L_j,S] = [L_j^\dagger,S] = 0$ where $\lambda \in \mathbb{R}$. As we will see in our numerical examples, however, the dynamics of a single quantum trajectory over a finite-time can provide clear evidence of Eq. (26).

Physically, the traceless modes which Eq. (26) implies represent information and coherences between symmetry subspaces which are protected from the system's interaction with the environment. These modes play an important role in the emergence of non-stationarity in open systems, [10], quantum synchronisation [17,18] and quantum information processing [46].

In the following section we will illustrate dissipative freezing in a number of numerical examples of open quantum systems. We will also observe the exceptions we have just described.

# 3 Results and Discussion

## 3.1 Numerical Details

For all of the following examples, we evolve our open system using the first-order trajectories routine described in Section 2.2. To minimise any incurred errors in the trajectory routine, we set $\Delta t \omega = 0.0025$ throughout, where $\omega$ is the central energy scale in the Hamiltonian and will be specified for each example. In terms of initial states, we will opt for two different types depending on what we wish to demonstrate

- I. Equal Weight: The initial state is initialised across two selected symmetry subspaces $\alpha_1$ and $\alpha_2$. Its weight in each subspace is the same and, within each subspace, it is an equal superposition of all basis states

$$|\psi(0)\rangle = \frac{1}{\sqrt{2}} \left( \sum_{\beta=1}^{d_{\alpha_1}} \frac{1}{\sqrt{d_{\alpha_1}}} |\alpha_1,\beta_1\rangle + \sum_{\beta=1}^{d_{\alpha_2}} \frac{1}{\sqrt{d_{\alpha_2}}} |\alpha_2,\beta_i\rangle \right), \tag{27}$$

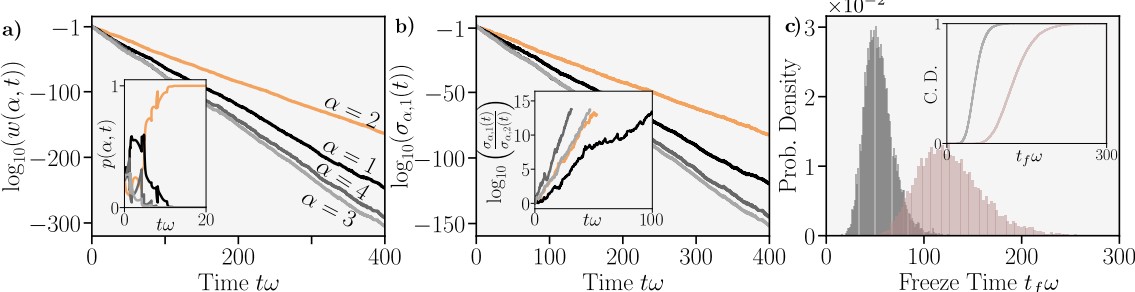

Figure 2: Freezing for a 'random' Liouvillian constructed from a random block-diagonal Hamiltonian $H$ and a single random block-diagonal jump operator $L_1$. There are four symmetry sectors, each of dimension 4 and, in each diagonal block, the Hamiltonian $H$ consists of a random Hermitian matrix scaled by a factor of $\omega$ whilst the single jump operator consists of a random operator drawn from the complex Ginibre ensemble [47]. We set $\gamma_1 = 4\omega$. For plots a-b the initial state is identical and of the form of Eq. (28) whilst for plot c it is of the form of Eq. (27) with the two chosen subspaces $\alpha_1$ and $\alpha_2$ being arbitrary. a) Logarithm of the unnormalised weight of the wavefunction in each subspace versus time for a single trajectory. Inset) Normalised weight of the wavefunction versus time. b) Logarithm of the largest singular values of the matrix $A_\alpha(t)$ in each subspace $\alpha$ for the same single trajectory. Inset) Logarithm of the ratio of the largest and second largest singular value in each subspace — values are not plotted once they can no longer be accurately determined. c) Probability density function for the individual freeze-times $t_f \omega$ for $10^4$ trajectories. The two different colours correspond to two independent realisations of the random Liouvillian — the initial state remains the same between them. Inset) Cumulative density function for the freeze time. We set $\epsilon = 10^{-6}$.

- II. Random: In each symmetry sector the initial state is set to a randomly chosen unit vector,

$$|\psi(0)\rangle = \frac{1}{\sqrt{D}} \sum_{\alpha=1}^{D} |\alpha, \beta_\alpha\rangle \,, \tag{28}$$

with $\beta_\alpha$ a randomly chosen integer in the range $(1, d_\alpha)$.

In our numerics, as per the discussion in Section 2.4, we will also be calculating the quantity $\tilde{p}(\alpha_1, t)\tilde{p}(\alpha_2, t)$ and defining freezing as having occurred when it falls below a threshold value $\epsilon \ll 1$. We will specify the value of $\epsilon$ used for the given example and demonstrate the bound derived in Eq. (24) as well as other properties that the freeze-time obeys.

## 3.2 Example: Random Matrices

As our first example, we consider the Lindblad equation, as in Eq. (1), with Hamiltonian and Lindblad operators which are block-diagonal random matrices — i.e. the off-diagonal blocks are set to 0 and the diagonal blocks of both the Hamiltonian and Lindblad operators consist of matrices drawn from the space of random Hermitian matrices and the complex Ginibre ensemble [47] respectively. This open system is then guaranteed to possess a strong symmetry in the form of an operator which is a different scalar multiple of the identity in each block.

In Fig. 2a-b we plot the evolution of the weights (both normalised and un-normalised) of a single trajectory in each symmetry subspace, along with the corresponding leading singular value of the matrix $A_\alpha(t)$ defined in Eq. (12) — dropping the $i$ superscript here as we are focused solely on a single trajectory. We use a logarithmic $y$-axis in order to expose the

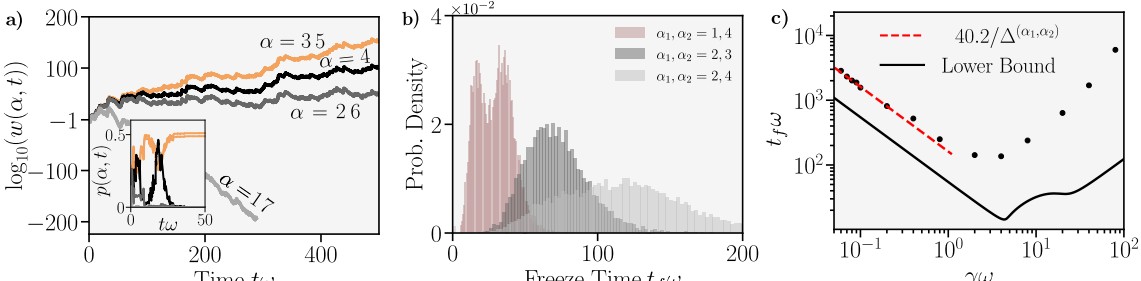

Figure 3: Freezing for coupled spin 3/2 particles governed by the GSKL equation with the Hamiltonian defined in Eq. (29) and a single jump operator $L = s_a^z$. We set $\gamma = 3\omega$. The index $\alpha = 1, 2, ..7$ refers to subspaces of total $z$ magnetisation $-3/2, -1, ..., 1, 3/2$ respectively. a) Logarithm of the un-normalised weight of a single trajectory — initialised in the form of Eq. (28) — versus time in each of the magnetisation subspaces. Inset) Normalised weight for $t\omega \leq 50$. b) Probability density of the freeze-time $t$ for $10^4$ trajectories and for trajectories initialised in the form of Eq. (27) in the indicated subspaces with $\gamma = 3\omega$. c) Average freeze-time (black dots) vs $\gamma\omega$ on a log-log scale for trajectories initialised in the form of Eq. (27) with $\alpha_1 = 2$ and $\alpha_2 = 3$ respectively. Red dotted line shows the function $40.2/\Delta^{(\alpha_1,\alpha_2)}$ whilst the black solid line shows the lower bound calculated via Eq. (24). We set $\epsilon = 10^{-6}$.

exponential growth of these quantities. Freezing occurs into the subspace which has the highest overall growth rate and the normalised weight in this subspace saturates to unity in the long-time limit. The exponential growth in the unnormalised weights coincides with an exponential growth of the leading singular values of the evolution matrix $A_\alpha(t)$ in each subspace $\alpha$. We also observe that the ratio of the largest and second largest singular values associated with this matrix is increasing at an exponential rate, justifying the approximation in Eq. (13). This growth is intrinsically tied to the initial state being propagated under an increasingly long-product of matrices.

In Fig. 2c we then plot histograms of the freeze times when measured over $10^4$ trajectories for two different realisations of the random Liouvillian — where a given realisation is constructed by drawing the block-diagonal Hamiltonian and jump operator from their respective ensembles. The initial state is defined in Eq. (27). The two histograms of freeze-times display qualitatively different distributions even though the matrices are drawn from the same underlying ensemble both times. This is due to the fact that the freeze-time will be strongly controlled by the spectrum of $(\hat{\mathcal{L}})_{\alpha_1,\alpha_2}$, which dictates the dynamics of coherences between the $\alpha_1$ and $\alpha_2$ subspaces. As per Eq. (23), the system cannot freeze until all these off-diagonal modes have died away sufficiently. The longest-lived of these modes will thus play a crucial factor in the freezing statistics and, in this example, these modes and their eigenvalues will clearly change for each draw of the block-matrices from their ensembles. In the next example we will consider a non-random, more physical Liouvillian, allowing us to straightforwardly calculate the gap $\Delta^{(\alpha_1,\alpha_2)}$ and directly correlate it with the corresponding freeze-times and our lower bound.

## 3.3 Example: Coupled Spin 3/2 Particles

For this example we move onto a more physical setup consisting of two coupled spin 3/2 particles — $a$ and $b$ — with particle $a$ undergoing continuous dephasing along the $z$ axis. The

local Hilbert space dimension is 4 and the Hamiltonian reads

$$H = \omega(s_a^x s_b^x + s_a^y s_b^y + s_a^z s_b^z), \tag{29}$$

where $s_{a/b}^\alpha$ is the spin-3/2 operator acting in the $\alpha = x, y, z$ direction on either spin $a$ or $b$. The dephasing is described by the single jump operator $L = s_a^z$.

This system possesses a strong symmetry in the form of the total $z$ magnetisation $A = s_a^z + s_b^z$ and thus the Hamiltonian and jump-operator can be simultaneously block-diagonalised into seven subspaces of dimensions $1, 2, 3, 4, 3, 2, 1$ and with respective total $z$ magnetisations $-3/2, -1, -1/2, 0, 1/2, 1$ and $3/2$. Whilst there are seven subspaces, we should not expect freezing between all of them. Specifically, the pairs of subspaces with the same absolute magnetisation are 'similar' (as per the definition in Eq. (25)), preventing freezing. We emphasize that there are no non-decaying traceless modes of the Liouvillian here and Eq. (10) holds for these pair of subspaces even though dissipative freezing does not occur. Dynamically, this follows from the fact that, apart from the initial state, the only difference in the dynamics in the two subspaces is that when a jump occurs the coefficients of the trajectory in the negative magnetisation subspace pick up a factor of $e^{i\pi}$ compared to the positive one. After a certain time, the total phase picked up will thus be $n\pi$, where $n$ is the number of jumps. The summation in the expression $\sum_i c_{\alpha',\beta'}^{(i)}(t) c_{\alpha'',\beta''}^{(i)}(t)$ can thus be split into two, one involving trajectories where $n$ is odd and one where $n$ is even. In the long-time limit, the probability that $n$ is even or odd should be equal and so the two terms should be identical, aside from a scale factor of $e^{i\pi}$, which causes them to cancel out and the ensemble-average to equal to 0 without freezing occurring.

In Fig. 3 we plot the evolution of trajectories in this system and illustrate these results, observing freezing between subspaces of different absolute magnetisation due to the distinct exponential growth of the matrices $A_\alpha(t)$ whilst observing an absence of freezing in subspaces with magnetisation of opposite sign due to the similarity of the subspaces. We then calculate the freezing times for trajectories initialised in a superposition of two different symmetry sectors. We observe distinctive distributions for each symmetry sector, indicating that they are specific to the form of the matrices in those sectors. We select the $\alpha_1 = 2$ and $\alpha_2 = 3$ sectors and analyse this further, varying $\gamma$ and computing the average freezing time, the inter-sector spectral gap $\Delta^{(\alpha_1, \alpha_2)}$ associated with these sectors and the lower bound on the freezing time as per Eq.(24).

We find that our bound is obeyed for the full range of $\gamma$ as expected and, whilst not being tight (which can be attributed to our application of the triangle inequality to a very large sum of complex numbers), it closely mimics the functional form of the freezing-times and demonstrates well their close relationship with the inter-sector spectral gap. For small values of $\gamma$ we observe proportionality between the inverse of the inter-sector spectral gap and the freezing time. This can be understood from perturbation theory [17], with the real parts of the eigenvalues of the Liouvillian's eigenmodes all scaling proportionally (with the same proportionality constant) with $\gamma$ for small $\gamma$. The inverse of these eigenvalues essentially corresponds to the timescales of the system and so the freeze-time will scale as the inverse of $\gamma$ and the inverse of the spectral gap.

## 3.4 Example: Lossy Bosonic Chain

For our final example we consider a periodic boundary chain of $L$ sites hosting non-interacting bosons collectively coupled to a cavity. In momentum space, the Hamiltonian reads

$$H = \omega a^\dagger a - g(a + a^\dagger) \sum_k b_k^\dagger b_{(k+\pi) \bmod 2\pi} - 2J \sum_k \cos(k) n_k, \tag{30}$$

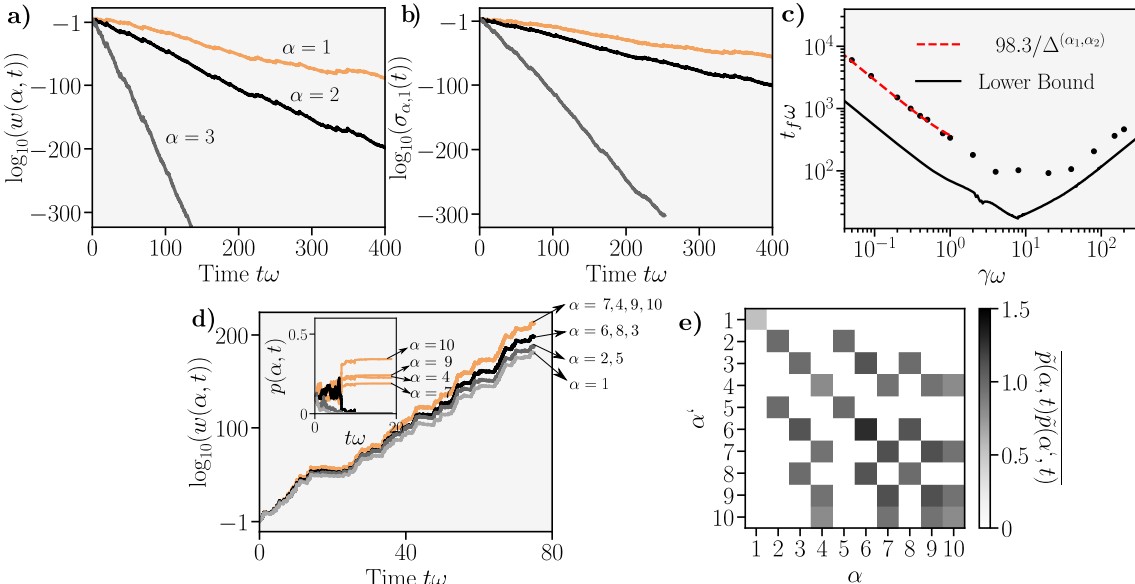

Figure 4: Freezing dynamics for non-interacting bosons collectively coupled to a single-mode electromagnetic field. The Hamiltonian is defined in Eq. (30) and there is a single jump operator $L = \sqrt{\gamma}a$. We set $\gamma = 5\omega$, $g = 2\omega$, $J = 2\omega$ and the system is always initialised in an equal superposition of random states in each symmetry subspace. The index $\alpha$ runs over the valid momentum tuples $(s_1, s_2, ..., s_{L/2})$ which specify the symmetry subspaces, the explicit relationship between $\alpha$ and these tuples is detailed in appendix A. We enforce a maximum photon number in our simulations of $n_{max} = 5$. In plots a-c) we set $L = 4$ whilst in plots d-e) we set $L = 6$. a) Logarithm of the weight of a single trajectory in each of the strong symmetry subspaces verus time, initial state is of the form of Eq. (28). b) Logarithm of the largest singular value of $A_\alpha(t)$ in each subspace versus time, initial state is of the form of Eq. (28). c) Average freeze-out time on a log-log plot for an initial state of the form of Eq. (27) with $\alpha_1 = 1$ and $\alpha_2 = 3$. Red dashed curve shows $98.3/\Delta^{(\alpha_1, \alpha_2)}$ whilst black solid line corresponds to the lower bound in Eq. (24). We set $\epsilon = 10^{-8}$. d) Logarithm of the weight of a single trajectory in each of the strong symmetry subspaces versus time, initial state is of the form of Eq. (28) Inset) Dynamics of the normalised weight for $t\omega \le 40$. e) Absolute values of the elements of the matrix $\tilde{p}(\alpha, t)\tilde{p}(\alpha', t)$ for $t\omega = 250$ and averaged over $10^3$ trajectories, initial state is the same for all realisations and of the form of Eq. (28).

where $a$ and $a^\dagger$, respectively annihilate and create photons in the cavity whilst $b_k$ and $b_k^\dagger$ respectively annihilate and create bosons in the chain with momenta $k \in \{\frac{2\pi}{L}, \frac{4\pi}{L}, ..., \frac{2\pi L}{L}\}$. We also define $n_k$ as the boson number operator for momentum $k$, i.e. $n_k = b_k^\dagger b_k$. Photon loss is introduced into the cavity via a single jump operator $L = a$ and we fix ourselves to half-filling, i.e. the total number of bosons is always $L/2$. The master equation formed from these operators was studied in Ref. [26] in the context of dissipative phase transitions of a cavity-immersed quantum gas. Importantly this setup possesses a series of $L/2$ commuting strong symmetry operators of the form $S_k = n_k + n_{k+\pi}$ with $k \le \pi$.

In Ref. [26] it was observed that, for the initial states considered, the individual trajectories in this system always freeze to a symmetry subspace. Here we unravel this phenomenon of dissipative freezing using the intuition we have built. Notably we also describe how, beyond a certain system size and for certain initial states, dissipative freezing is not guaranteed due to the fact that the inter-sector gap closes for certain pairs of subspaces, i.e $\Delta^{(\alpha_1, \alpha_2)} \to 0$.

First, as the strong symmetry operators all commute, there clearly exists a simultaneous block-decomposition of the Hamiltonian and jump-operator. Specifically, the corresponding subspaces or blocks can be indexed by the unique tuples $(s_1, s_2, ..., s_{L/2})$ where $s_i$ corresponds to the number of bosons in momentum modes $k = \frac{2\pi i}{L}$ and $k = \frac{2\pi i}{L} + \pi$, i.e. $s_i = \langle n_k + n_{k+\pi} \rangle$. If the total number of bosons is $N$ then it follows that the number of unique tuples (or subspaces) is $D = \frac{(N+L/2-1)!}{N!(L/2-1)!}$ [48] which grows exponentially with $L$ for $N = L/2$.

An initial state initialised in a superposition of states with different tuples is therefore expected to undergo the phenomenon of dissipative freezing — which is exactly what we observe in Fig. 4a-b for a small $L = 4$ site chain with the initial state having a non-zero weight in all subspaces. The characteristic exponential growth in the weights and singular values is observed just like in our earlier examples.

In Fig. 4c we again find the freezing times obey the bound we derived in Eq. (24). Notably as $\gamma$ becomes larger this bound becomes tighter. This can be understood from the fact that for increasing $\gamma$ the Hamiltonian's role in the dynamics becomes increasingly neglibible and the trajectory propagator tends towards a matrix whose elements are always positive real values. The triangle inequality used to derive our bound thus becomes tighter as $\gamma$ increases and so does our lower-bound on the freeze time. It is difficult, however, to reach a numerical regime where this bound saturates as it requires using a very large value of $\gamma$ (which requires a smaller $\Delta t$ to maintain accuracy), reaching exceptionally long timescales and maintaining signficant precision in the vanishingly small value of $\tilde{p}^{(i)}(\alpha_1, t)\tilde{p}^{(i)}(\alpha_2, t)$. In Fig. 4c we also observe direct proportionality between the freeze-time and the inverse of the inter-sector gap for small $\gamma$ — the regime in which $\gamma$ acts as a rescaling of the system's timescales.

It might be natural to assume that freezing will occur in this Bosonic setup for all system sizes. In Fig. 4c-d we treat a system for $L = 6$ and observe that this is not the case, with freezing only occurring between certain subspaces. The origin of this lack of freezing are the traceless modes of the Liouvillian, $\Delta^{(\alpha_1, \alpha_2)} \to 0$, for $L = 6$ and certain pairs of subspaces. In our numerics we also observe that these modes also appear for $L > 6$ and independently of the maximum number of photons that we enforce. For $L = 6$ these modes connect the symmetry sectors where $s_1 + s_2$ is the same, which likely is related to the fact that the $\cos(k)$ term in the Hamiltonian only differs by a minus sign for the two *separate* (i.e. not simply related by a shift of $+\pi$) momentum modes $k = 2\pi/3$ and $k' = 4\pi/3$ which correspond to $s_1$ and $s_2$ respectively. When $L \geq 6$ we can always identify such pairs of modes where $\cos(2\pi i/L) = -\cos(2\pi j/L)$ and $i, j \in 1, 2, ..., L/2$. For $L = 4$ this is not possible.

These traceless modes are clearly manifest as non-zero off-diagonal elements in the ensemble-averaged matrix $\overline{\tilde{p}(\alpha_1, t)\tilde{p}(\alpha_2, t)}$ plotted in Fig. 4e. Following Eq. (23), for sufficiently long times we can be confident this matrix provides a clear picture of where the traceless non-decaying modes of the system reside (i.e. where the inter-sector spectral gap is 0). Even a single trajectory initialised across all subspaces, however, allows us to infer this information. If there are traceless modes and the freezing time diverges, then any given trajectory should have a negligible probability of freezing on any finite time-scale.

Figure 4d explicitly demonstrates this information being inferred at the single-trajectory level: all pairs of subspaces which share a traceless mode have weights of the same order of magnitude for all time, in direct agreement with the ensemble averaged picture provided by Fig. 4e. This single-trajectory picture thus immediately points to a steady state which is not block-diagonal for initial states containing coherences which overlap with these modes. The freezing time thus diverges but, as opposed to the previous setups, this happens for all finite $\gamma\omega$ and does not require it to to tend to 0 or $\infty$.

Importantly, our results in this paper show we do not need to wait until $t \to \infty$ to infer this lack of freezing at the single trajectory level. Figure 4 pictures a clear manifestation of these traceless modes after a finite period of time, with the probabilities in the associated

subspaces saturating to a constant, non-zero non-unity value. Given the identification of a strong symmetry and the corresponding symmetry subspaces, the ability to utilise a single trajectory to detect the existence of these traceless modes is a much more computationally efficient method than having to resort to ensemble averaging or memory-intensive calculations in the superoperator picture.

We note that these traceless modes and the resultant absence of freezing between certain subspaces was not observed in Ref. [26] as the initial states used did not span the necessary subspaces to have an overlap with these modes.

## 4 Conclusion and Outlook

In this work we have provided significant mathematical and physical insights into understanding how the recently-observed phenomenon of dissipative freezing is a general consequence of the presence of a strong symmetry in an open system. The associated symmetry-breaking in the trajectories picture is the way the system ensures symmetry is preserved at the ensemble level. We have also introduced a range of examples where dissipative freezing can be directly observed and quantified.

In these examples, we showed how the freezing statistics are dependent on the lowest lying eigenvalue in the 'off-diagonal' superoperator subspaces. When the real part of this eigenvalue vanishes, then the Liouvillian possesses traceless non-decaying modes, immediately implying an absence of dissipative freezing for any single given trajectory. We have thus identified a computationally efficient method for identifying the existence of such modes.

We envisage a number of future avenues of research stemming from this work. Firstly, identifying whether further exceptions to this phenomenon are possible or whether those identified in this work — traceless non-decaying modes and similar subspaces — are the only possible exceptions, would greatly further our understanding of the effect of symmetries on trajectories in open quantum systems.

Secondly, in this work we have adopted the approach of being as general as possible and making no assumptions about the microscopic details of the Liouvillian other than its possession of a strong symmetry. We anticipate that by making stronger assumptions about the details of the system, the bound in Eq. (24) can be improved and the general dependence of the the freezing properties of the system on its spectral statistics can be quantified further — leading to novel ways to determine the spectrum of open systems from within the memory efficient trajectories formalism.

Lastly, and by no means least, an important further pursuit would be to quantify the role that symmetry-breaking perturbations play in the breakdown of dissipative freezing. Such perturbations would lead to an opening of the Liouvillian's gap in the full superoperator space and quantifying this through the freezing statistics could provide new insights into dissipative phase transitions in open systems.

## Acknowledgements

C.S.M. acknowledges that the project which gave rise to these results received the support of a fellowship from "la Caixa" foundation (ID 100010434) and from European Union's Horizon 2020 research and innovation programme under the Marie Skłodowska-Curie Grant Agreement No. 847648, with fellowship code LCF/BQ/P120/11760026, and financial support from the Proyecto Sinérgico CAM 2020 Y2020/TCS-6545 (NanoQuCo-CM). DJ and JT acknowledge funding from EPSRC grant EP/P009565/1. DJ also acknowledges funding from the Cluster

of Excellence 'Advanced Imaging of Matter' of the Deutsche Forschungsgemeinschaft (DFG) - EXC 2056 - project ID 390715994. We are also grateful to Berislav Buča for discussions on dissipative freezing and for suggesting the coupled model.

## Appendix A: Indexing the subspaces of the lossy bosonic chain

In Table 1 we tabulate the relationship between the single subspace index $\alpha$ we adopt in the figures and the unique tuples $(s_1, s_2, ..., s_{L/2})$ which explicitly define the strong symmetry subspaces for the lossy bosonic chain in Section 3.4.

Table 1: Relationship between the subspace index $\alpha$ we adopt in Figure 4 and the momentum tuples $(s_1, s_2, ...., s_{L/2})$ which define the symmetry subspaces in the lossy bosonic chain. Left table is for $L = 4$ and right is for $L = 6$.

**L = 4:**

| $(s_1, s_2)$ | $\alpha$ |
|---|---|
| (0,2) | 1 |
| (1,1) | 2 |
| (2,0) | 3 |

**L = 6:**

| $(s_1, s_2, s_3)$ | $\alpha$ |
|---|---|
| (0,0, 3) | 1 |
| (0,1,2) | 2 |
| (0,2, 1) | 3 |
| (0,3,0) | 4 |
| (1,0,2) | 5 |
| (1,1,1) | 6 |
| (1,2,0) | 7 |
| (2,0,1) | 8 |
| (2,1,0) | 9 |
| (3,0,0) | 10 |

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
