# Peer review of "On the generality of symmetry breaking and dissipative freezing in quantum trajectories"

_SciPost Physics, doi:SciPost Phys. Core 6, 004 (2023)_

## Round 1 · Referee Report · Marco Cattaneo (Referee 1) · 2022-5-16

Strengths

1- It introduces a satisfactory explanation of why dissipative freezing occurs.
2- It presents some nice examples that corroborate this explanation.

Weaknesses

1- The clarity of some concepts and sentences is occasionally not so high.

Report

The authors present a number of arguments for the emergence of dissipative freezing in quantum trajectories. Dissipative freezing is the fact that, at the level of a single quantum trajectory, the evolved state is quite often found at infinite time in a single subspace only of a strong symmetry. They also show how and why this phenomenon is broken.

Overall, I think the paper is very good and the explanation for the emergence of dissipative freezing is convincing and interesting, especially for the argument on long-products of matrices. The examples are also really nice and pedagogical.

I must say, however, that sometimes I struggled to follow the text and I believe some clarifications are necessary. For instance, I feel the presentation is too discursive in some sections, while some more formulas wouldn't hurt.

I would like the authors to address the following comments:
1- I do not exactly understand where the time derivatives in Eq.(10) (for m=1,2,...) come from. Anyway, isn't the equation for m=0 sufficient to make the authors' point? If nothing weird happens and the time derivative commutes with the limit, we are just saying that the time derivatives at infinite time of some functions that are asymptotically zero (for t -> infty) are zero. Isn't this kind of trivial? So, aren't the time derivatives redundant here?
2- Right after eq. (10), the authors say "at least one of coefficients in the product... will always be zero". For alpha \neq alpha', aren't all the coefficients zero?
3- I have a curiosity about the definition of "traceless non-decaying eigenmode" of the Liouvillian. Apparently, according to the authors this means that there is some non-decaying coherence between different symmetry eigenspaces. Isn't it possible to have a traceless non-decaying eigenmode within a single symmetry subspace (oscillating coherences with the same symmetry value)?
4- The meaning of the last sentences in Section 2.3 "Recovering ... Steady State" is a bit obscure. Given an open quantum system, why shall we be allowed to add symmetry subspaces? Please clarify this or provide some examples.
5- Please provide a more clear definition of "growth rate" in the section about Long-products of matrices. A formula would help the reader to better grasp this concept.
6- How is the ergodic time average defined in the references [41,42]? Is there some kind of time window in the average, or is it 1/t int_0^t for t that goes to infinity? Please specify this.
7- I miss something in the discussion of the result in Ref. [41]. How is it connected to Eq. (10)? Is it because ergodicity for a single trajectory would exclude the possibility for this trajectory to reach a state in a single symmetry subspace only? What if the trajectory passes ergodically through all the different states in different symmetry sectors, and the time average yields the ensemble average? Maybe this is not really clear, but please, clarify this point through a more extensive discussion.
8- It is really difficult to follow the last sentences of the subsection on ergodicity. Please, try to express with some formulas the final sentence starting with "If the probability of freezing into...". I believe it's much easier for the reader to understand your message if there are some clear and readable equations to look at.
9- In section 2.4 two different kinds of exception are discussed. Am I wrong or the first kind is related to the absence of dissipative freezing despite all the steady states can be written as in Eq. (8) (at least in almost all scenarios), while the second one is due to the failure of Eq.(8)? I think highlighting this difference could be good.
10- Please introduce the "freeze-time" in Section 3.1 through a proper equation.
11- Same as for the gap Delta in the following example. Write down in a visible equation what you are talking about.

Finally a few typos: pag. 2 last paragraph, "allows us (to) identify"; the caption of Fig. 2, "unormalised"; the name of the first author of Ref. [42] is misspelled.

Requested changes

Please consider the above points.

  • validity: high
  • significance: top
  • originality: top
  • clarity: ok
  • formatting: excellent
  • grammar: excellent

Author:  Joseph Tindall  on 2022-07-11  [id 2649]

(in reply to Report 1 by Marco Cattaneo on 2022-05-16)

1) The time-derivatives were in place to emphasize that the expression is `steady’ i.e. that the ensemble average of coefficients is 0 and unchanging in the long-time limit. The referee is correct, however, that this is superfluous and $m=0$ is sufficient because if the limit $t \rightarrow \infty$ converges asymptotically for $m=0$ then its derivative must vanish.

2) For a given trajectory (i.e. specific $i$) if freezing has occurred then there is only a single $\alpha$ for which the coefficients $c^{(i)}_{\alpha, \beta}(t)$ are not zero. Hence, for a given $i$, the product of coefficients for two different values of $\alpha$ is zero. We have now made this clear.

3) Yes, we did not intend our definition to imply you cannot have traceless non-decaying modes within a single symmetry subspace. We have renamed these as `inter-sector traceless modes’ in order to avoid suggesting these are the only possible traceless, non-decaying modes that can appear in a Liouvillian’s spectrum.

4) One is always able to introduce a new symmetry subspace to the system by expanding their Hamiltonian and jump operators via a direct sum. Physically this would correspond to considering an additional, independent open system on top of the existing one.

5) We now provide a mathematical definition of this average growth rate.

6) This time average is defined via an average between times zero and infinity. We have specified this and improved the readability here as the sentence was confusing in its original form.

7) The connection we were implying is that establishing an ergodic theorem with a single trajectory across multiple symmetry sectors is not possible due to the problem being over constrained in analogy with Eq. (10). In the interests of avoiding discursive discussion we have removed the reference to Eq. (10) here.

8) We have introduced explicit formulas for the quantities mentioned at the end of Section 2.3. With the addition of more explicit formulas for the freeze-time and inter-sector spectral gap (see point 10), this should be easier to follow.

9) Yes, the first is a breakdown of freezing despite Eq. (8) holding whilst the second is a scenario in which Eq. (8) is no longer true. Section 2.5 has been reworded to make this clearer.

10 + 11) We have introduced an equation for the freeze time and inter-sector spectral gap. Our expression for the freeze-time now follows from a new section where we derive a lower-bound on the timescale of freezing.

---

## Round 1 · Referee Report · Anonymous (Referee 2) · 2022-5-19

Strengths

- Quite accessible introduction to the topic

Weaknesses

1- The paper lacks precision at several instances.
2- Although the statements about dissipative freezing are formulated in a mathematical language, the arguments are rather handwaving and no quantitative aspects of dissipative freezing, i.e. no boundaries, time scales etc. are discussed.
3- To me the paper does not provide much original content compared to previous publications on dissipative freezing. To be candid, what the authors present in Sec. 1 and 2 was the picture that was more or less communicated in previous publications [24-26].

Report

The manuscript is quite accessible and explains the phenomenon of dissipative freezing in simple terms. This phenomenon, which describes the time-dependent collapse of a wave function into one of the symmetry sectors of the Liouvillian generator of the dynamics was observed recently. Very colloquially, the reason for this phenomenon to occur is the one-way type stochastic evolution of each unravelling, which will "loose" contributions from different symmetry sectors over time but, due to the symmetry, is unable to restore them.

In summary, I think the paper is correct and well written and therefore deserves publication. However, in my personal opinion it lacks the originality and the quantitative precision that would be necessary to publish in SciPost Physics. I therefore think that, after consideration of the comments below, I may be published in SciPost Physics Core instead.

Details to weakness 2:
The authors' general argument for the occurrence of dissipative freezing is based on Eq. (10). However, this implies that all derivatives of this equation go to zero in finite time. I don't see a clear reason why this would be the case. All statements based on the stationary density matrix are asymptotic (i.e. t-> infty and N -> infty). The order of the two limits is important as well, since if we want to describe the density matrix entries at all times, one has to take N-> infty first and then t-> infty. Then number of the degrees of freedom, which need to fulfil the constraints grows as fast as the number of constraints. Therefore I am not convinced by the authors' argument.
Even if I was convinced, the lack of quantitative statements about the process persists.

Lack of precision:

- page 4: "D trace unity hermitian steady states", I don't understand what is meant by this expression. There is an infinite number of possible stationary, each one depending on the structure of the initial state. The basis of stationary states is at least D-dimensional.

- page 4: "We emphasize that this unravelling of the GSKL equation is not just a mathematical formalism:..." The authors should be more precise at this point. First of all, one needs to distinguish unravellings with hermitian jump operators from unravellings with nonhermitian jump operators. The latter is per se not a continuous measurement process. Furthermore, even for hermitian jump operators one can easily choose an unravelling that has no measurement counterpart.

- page 5: Definition of dissipative freezing: The authors say the process happens generically but also mention that there are few exceptions. Neither the term generic nor the number of "few" is explained here. Is it clear that the two exceptions which are discussed later in the text are the only exceptions? This doesn't read very convincing.

- Eq. (15) can't be correct. If P_1 and P_2 project onto orthogonal subspaces then P_1HP_1=P_2HP_2 implies that the Hamiltonian must be zero in both subspaces. If P_1|psi>=|psi> then P_2|psi>=0. I guess what the authors mean is that in both subspaces the Hamiltonian has the same matrix structure (modulo a unitary transformation since I can always choose a different subsystem basis).

  • validity: good
  • significance: good
  • originality: good
  • clarity: ok
  • formatting: excellent
  • grammar: excellent

Author:  Joseph Tindall  on 2022-07-11  [id 2648]

(in reply to Report 2 on 2022-05-19)

Response to main criticism:

Originality: We disagree that References [24-26] communicated the picture we provide in Section 1 and 2 of our work. All these references considered a single microscopic model and numerically observed dissipative freezing (DF) (with Ref. 24 the only one to provide a mathematical proof and this was specific to the exceptional situation H = L). We instead make no assumptions about the details of the Liouvillians—other than its possession of a strong symmetry—and provide significant mathematical and physical insight into why freezing will emerge. We also identify a direct connection between rigorous mathematical results on ergodicity [40-42] in open systems and dissipative freezing – something which Refs. [24-26] all missed. We did not intend our work to constitute a rigorous mathematical proof of DF. The aim of this paper is, instead, to provide an insightful, comprehensive, mathematical and physical understanding of its origin, complemented with a range of concrete numerical examples. Alongside the intuition and comprehension it provides, such a perspective has allowed us to develop a number of original results without needing to rely on heavy mathematical artillery: exceptions to DF, a lower bound on the timescale on which DF occurs and a computationally efficient method for identifying traceless non-decaying modes in a Liouvillian. In our revised version of the manuscript we have made these points much clearer.

Precision: Both referees comments have significantly helped us improve the precision of the manuscript and the changes made are listed at the end of the document. These changes include mathematical definitions for certain quantities, rectifying grammatical and mathematical errors, avoiding discursive discussions, providing several elaborations when requested and deriving a rigorous lower bound on the time-scale of dissipative freezing – which we demonstrate in our numerics. We believe that these changes have significantly improved the clarity and precision of the manuscript (see list of changes to manuscript).

Responses (to referee comments under `Lack of Precision’):

1) The basis of steady states is at least D dimensional, this is now stated explicitly.

2) We have provided additional discussion and references about the relationship between a stochastic unravelling of the GSKL equation and a continuous measurement process in an open system. We note that non-Hermitian jump operators can result in a physical unravelling – for instance via photon counting in the environment.

3) The ‘Traceless’ mode exception completely covers the situation in which the inter-sector spectral gap vanishes, leading to Eq. (10) not being valid and freezing not occurring. The `Similar Symmetry Subspaces’ exception we detail is the only instance we are aware of where Eq. (10) can hold and the system does not undergo dissipative freezing. It is possible other examples where the structure of the two subspaces is sufficiently similar to prevent freezing exist. We have made this explicit, along with the fact we are, however, unaware of any such examples.

4) We thank the referee for spotting this error—we meant to say the projected Hamiltonian has the same form when focussing on the diagonal block where it is non-zero and have now made this mathematically precise, including the part about the unitary transform (change of basis).

---

## Round 2 · Referee Report · Marco Cattaneo (Referee 1) · 2022-7-11

Report
The authors have modified the text according to my suggestions and I believe the presentation has considerably improved. In particular, the new section 2.4 provides us with a quantitative description of dissipative freezing that conveys the authors' message in a more clear way.
I would like to recommend the manuscript for publication, once the following minor typos have been fixed: 1) In eq.(10) the authors forgot to remove the text $m \in $ ... 2) In eqs. (21), (22) and the text below them there is a missing $'$ in $\beta$ in the definition of $\rho_R$.

Author: Joseph Tindall on 2022-08-04 [id 2712]
(in reply to Report 1 by Marco Cattaneo on 2022-07-11)We are pleased the referee recommends our work for publication and we have now fixed the typos they have pointed out.

---

## Round 2 · Author Response

Dear Editor and Referees,
We thank both referees for their thorough comments and reading of the manuscript. Following these comments, we have revised the manuscript substantially and believe it merits publication in Scipost Physics. Below, we first respond to the main criticisms of Referee 2, we then provided itemised responses to the individual comments of the referees. A list of the changes we have made to the manuscript is also provided for the resubmission.
The authors, Joseph Tindall, Dieter Jaksch and Carlos-Sanchez Munoz
Main Criticism from Referee 2 (Lack of precision and originality):
Here we address the referee’s criticism about the originality and precision of the manuscript.
Originality: We disagree that References [24-26] communicated the picture we provide in Section 1 and 2 of our work. All these references considered a single microscopic model and numerically observed dissipative freezing (DF) (with Ref. 24 the only one to provide a mathematical proof and this was specific to the exceptional situation H = L). We instead make no assumptions about the details of the Liouvillians—other than its possession of a strong symmetry—and provide significant mathematical and physical insight into why freezing will emerge. We also identify a direct connection between rigorous mathematical results on ergodicity [40-42] in open systems and dissipative freezing – something which Refs. [24-26] all missed. We did not intend our work to constitute a rigorous mathematical proof of DF. The aim of this paper is, instead, to provide an insightful, comprehensive, mathematical and physical understanding of its origin, complemented with a range of concrete numerical examples. Alongside the intuition and comprehension it provides, such a perspective has allowed us to develop a number of original results without needing to rely on heavy mathematical artillery: exceptions to DF, a lower bound on the timescale on which DF occurs and a computationally efficient method for identifying traceless non-decaying modes in a Liouvillian. In our revised version of the manuscript we have made these points much clearer.
Precision: Both referees comments have significantly helped us improve the precision of the manuscript and the changes made are listed at the end of the document. These changes include mathematical definitions for certain quantities, rectifying grammatical and mathematical errors, avoiding discursive discussions, providing several elaborations when requested and deriving a rigorous lower bound on the time-scale of dissipative freezing – which we demonstrate in our numerics. We believe that these changes have significantly improved the clarity and precision of the manuscript.
Report 1 Responses:
1) The time-derivatives were in place to emphasize that the expression is `steady’ i.e. that the ensemble average of coefficients is 0 and unchanging in the long-time limit. The referee is correct, however, that this is superfluous and $m=0$ is sufficient because if the limit $t \rightarrow \infty$ converges asymptotically for $m=0$ then its derivative must vanish.
2) For a given trajectory (i.e. specific $i$) if freezing has occurred then there is only a single α for which the coefficients $c_{\alpha, \beta}^{(i)}(t)$ are not zero. Hence, for a given $i$, the product of coefficients for two different values of α is zero. We have now made this clear.
3) Yes, we did not intend our definition to imply you cannot have traceless non-decaying modes within a single symmetry subspace. We have renamed these as `inter-sector traceless modes’ in order to avoid suggesting these are the only possible traceless, non-decaying modes that can appear in a Liouvillian’s spectrum.
4) One is always able to introduce a new symmetry subspace to the system by expanding their Hamiltonian and jump operators via a direct sum. Physically this would correspond to considering an additional, independent open system on top of the existing one.
5) We now provide a mathematical definition of this average growth rate.
6) This time average is defined via an average between times $0$ and $\infty$. We have specified this and improved the readability here as the sentence was confusing in its original form.
7) The connection we were implying is that establishing an ergodic theorem with a single trajectory across multiple symmetry sectors is not possible due to the problem being over constrained in analogy with Eq. (10). In the interests of avoiding discursive discussion we have removed the reference to Eq. (10) here.
8) We have introduced explicit formulas for the quantities mentioned at the end of Section 2.3. With the addition of more explicit formulas for the freeze-time and inter-sector spectral gap (see point 10), this should be easier to follow.
9) Yes, the first is a breakdown of freezing despite Eq. (8) holding whilst the second is a scenario in which Eq. (8) is no longer true. Section 2.5 has been reworded to make this clearer.
10 + 11) We have introduced an equation for the freeze time and inter-sector spectral gap. Our expression for the freeze-time now follows from a new section where we derive a lower-bound on the timescale of freezing.
Report 2 Responses (under `Lack of Precision’)
1) The basis of steady states is at least $D$ dimensional, this is now stated explicitly.
2) We have provided additional discussion and references about the relationship between a stochastic unravelling of the GSKL equation and a continuous measurement process in an open system. We note that non-Hermitian jump operators can result in a physical unravelling – for instance via photon counting in the environment.
3) The ‘Traceless’ mode exception completely covers the situation in which the inter-sector spectral gap vanishes, leading to Eq. (10) not being valid and freezing not occurring. The `Similar Symmetry Subspaces’ exception we detail is the only instance we are aware of where Eq. (10) can hold and the system does not undergo dissipative freezing. It is possible other examples where the structure of the two subspaces is sufficiently similar to prevent freezing exist. We have made this explicit, along with the fact we are, however, unaware of any such examples.
4) We thank the referee for spotting this error—we meant to say the projected Hamiltonian has the same form when focussing on the diagonal block where it is non-zero and have now made this mathematically precise, including the part about the unitary transform (change of basis).

---

## Round 2 · List of Changes

1) Introduction. We have mentioned the bound we derive on the freeze-time and improved the wording to make the contributions of the paper to the field clearer. 2) Section 2.1. We have clarified that we mean the basis of steady states is at least D dimensional. 3) Section 2.2: We have expanded on what we mean by ‘generically’ in the definition of dissipative freezing. Section 2.3: Recovering a Block Diagonal Steady State: The time derivatives have been removed from Eq. (10). The sentence following Eq. (10) has been made clearer. The discussion at the end of this subsection has been made more concise and clear. 4) Section 2.3: Long Products of Matrices: We have mathematically defined the average growth rate of the singular values of the propagation matrices in a given symmetry subspaces. 5) Section 2.3 Ergodicity and Quantum Systems Subject to Repeated Measurements: We have removed the discussion about Eq. (10) in this section to avoid being too discursive. We have mentioned that the time-average in the references is defined as between times $0$ and $\infty$. 6) New Section 2.4: The Timescales of Dissipative Freezing. This is a new section in which we derive a lower bound on the freezing time of trajectories. In this section we have also provided a mathematical definition for the inter-sector spectral gap, which plays a central role in the lower bound. 7) Section 2.5 (Previously Section 2.4): We have emphasized that Eqs. (8) and (10) are maintained in the first case whilst breaking down in the second, making the two exceptions very distinct. We have correlated the breakdown of freezing in terms of the bound derived in Sec 2.4. We have corrected a mistake in Eq. (15), including the unitary transform, and re-worded the sentences following to provide clarity. We have also made it clear that other exceptions involving similar subspaces may exist beyond Eq. (15). 8) Section 3.1: We have provided a new and clearer definition (the qualitative results remain the same as before) of the average freeze-time, connecting it to our new section 2.4 (on the timescales of freezing). We have mathematically defined the types of initial state used in the numerics to be precise. 9) Section 3.2-3.4 Numerical Results. We have redefined the freeze-time and updated the plots based on this (the qualitative physics is unchanged). We have plotted the lower bound on freeze-time in Figs 3c and 4c. We have significantly updated the discussions to reflect the earlier changes in the manuscript (e.g. new section and the earlier definition of quantities like the inter-sector spectral gap). This makes the discussion in this section much more precise. 10) Throughout: Grammatical errors pointed out by the referees have been fixed, included a naming error in the references.

---

## Round 3 · Referee Report · Anonymous (Referee 3) · 2022-8-9

Strengths

1. Well structured and accessible as only minimal knowledge of linear algebra is required.
2. Convincing examples demonstrate dissipative freezing and exceptions to it.

Weaknesses

1. Lacks precision and rigor in its arguments and conclusions

Report

Dissipative freezing corresponds to a recently observed property of the time-evolution of an open system with a strong symmetry to asymptotically project each quantum trajectory into a single subspace of the strong symmetry, thereby spontaneously breaking the latter at the level of individual trajectories.
The manuscript attempts to provide a broader view on this phenomenon that should be accessible to most physicists while also attempting to elevate the discussion previously constrained to specific examples to a more formal level. Unfortunately, the manuscript falls short of its ambitions, ending up as neither fish nor fowl.
The point where this becomes most painfully clear is the box on page 5, which is unsuitable as a mathematical definition but also provides little physical insight as no concise conditions for exceptions are provided.
The level of the analysis of dissipative freezing presented in the manuscript, nevertheless, significantly exceeds the existing literature. I am therefore generally inclined to recommend publication if the following minor points can be clarified/corrected or improved:

Requested changes

1. Content:

In the introduction, I would appreciate a paragraph on the physical relevance of dissipative freezing. It is discussed later, but in my opinion should also be emphasized in the introduction.

Below equation (14) there appears to be a typo as neither $\alpha$ nor $\alpha’$ are defined in this context. I therefore suggest to replace “is largest in $\alpha$ or $\alpha'$” by “is larger in $\alpha_1$ or $\alpha_2$”.

At the beginning of page 10, it doesn’t seem necessary, nor guaranteed by the definition of A that the steady state of each symmetry sector is unique. To be explicit, A might not split the Hilbert space in the maximal number of subspaces.

At the beginning of page 11 I fail to follow the argument
“if $\tilde{p}^{(i)}(\alpha_{1}, t)\tilde{p}^{(i)}(\alpha_{2}, t) \ll 1$ we have that one of $p^{(i)}(\alpha_{1},t)$ or $p^{(i)}(\alpha_{2}, t)$ will be on the order of $\epsilon^{2}$”
In this context it might also help to introduce the symbol $t_f$ for the freeze-time.

In section 2.5 $\Delta^{(\alpha_{1}, \alpha_{2})} = 0 \ \forall \alpha_{1} \neq \alpha_{2}$ should read $\Delta^{(\alpha_{1}, \alpha_{2})} \neq 0 \ \forall \alpha_{1} \neq \alpha_{2}$. However, as is explained in the following section on inter-sector traceless modes, even this statement is not generally valid.

Conversely in equation (26) one should have $\Delta^{(\alpha_{1}, \alpha_{2})} = 0$. On a more general note, this condition doesn’t seem very useful as, on the one hand, it is not being linked to any properties of the Hamiltonian or Lindblad operators. On the other hand, it is also computationally expensive to check. If possible I would appreciate if the authors code provide a statement to which extend equation (26) may be of use.

The argument on page 17 “If the trajectory is left to evolve over a time-scale sufficiently longer than that associated with the inverse of $\Delta^{(\alpha_{1}, \alpha_{2})}$, then any signatures of freezing should be apparent. If they are not then it is reasonable to assume freezing will not occur and traceless non-decaying modes are present.” is circular as an evolution to $t\sim 1/\Delta^{(\alpha_{1}, \alpha_{2})}$ excludes the case of traceless non-decaying modes characterized by $\Delta^{(\alpha_{1}, \alpha_{2})}$

2. Grammar/Spelling:

While the general level of grammar is very good, in its current form the manuscript is riddled with punctuation errors. Due to their huge number, I will not provide the detailed location, but rather the immediate context:

- far reaching implications which→ far-reaching implications, which

- model at hand an analytical → model at hand, an analytical

- numerical examples, from a range → numerical examples from a rang

- absense → absence

- dissipative freezing we provide a novel → dissipative freezing, we provide a novel

- system symmetries can no longer → system, symmetries can no longer

- also true, if there → also true if there

- Evan’s → Evans or Evans’

- Unravellings which are to higher order → Unravellings which are of a higher order

- symmetry subspace then the phenomenon → symmetry subspace, then the phenomenon

- off-diagonal block we will refer them as → off-diagonal block, we will refer to them as

- states in different subspaces then dissipative → states in different subspaces, then dissipative

- constraint independently → constraint, independently

- of the GSKL equation we can → of the GSKL equation, we can

- this make sense → this makes sense

- jump operators it follows from → jump operators, it follows from

- quantities values → quantities’ values

- the, randomly selected → the randomly selected

- freezing we now → freezing, we now

- our analysis we work → our analysis, we work

- block-diagonal then the Liouvillian → block-diagonal, the Liouvillian

- If this is true then → If this is true, then

- simlarity → similarity

- any, other forms of → any other forms of

- respresent → represent

- is protected from the system’s → are protected from the system’s

- examples we → examples, we

- routine we → routine, we

- As our first example we → As our first example, we

- Lindblad operators which are block → Lindblad operators, which are block

- and in this example these → and, in this example, these

- a certain time the total → a certain time, the total

- the long-time limit the probability → the long-time limit, the probability

- identical, aside → identical, aside

- Importantly this→ Importantly, this

- intuition we have built up → intuition we have built

- all commute there → all commute, therefor
- follows the number → follows that the number

- Notably as $\gamma$ → Notable, as $\gamma$

- is traceless modes → are the traceless modes

- for sufficiently long times we can → for sufficiently long times, we can

- diverges then any given → diverges, then any given

- are not then → are not, then

- much more computationally → a much more computationally

- vanishes then the Liouvillian → vanishes, then the Liouvillian

- posseses → possesses

- exceptions would → exceptions, would

- possesion → possession

- system the bound → system, the bound

- and then general dependence of the the freezing→ and then the general dependence of the freezing

- where the $c^{(i)}_{\alpha, \beta}(t)$ are the coefficients of the $i$th trajectory → where $c^{(i)}_{\alpha, \beta}(t)$ is the coefficient of the $i$th trajectory

- to ensemble average → to ensemble-average

- open systems) the more constraints → open systems), the more constraints

- qubit → qudit

  • validity: high
  • significance: high
  • originality: good
  • clarity: high
  • formatting: excellent
  • grammar: good

Author:  Joseph Tindall  on 2022-09-23  [id 2845]

(in reply to Report 1 on 2022-08-09)

We are thankful for the thorough reading and comments on the manuscript. We have made a number of changes based on their recommendations.

---

## Round 3 · Referee Report · Anonymous (Referee 4) · 2022-9-4

Strengths

1 - Well-structured and accessible manuscript, suitable for a wide readership.
2- Timely topic
3- Nice set of examples to help understanding and giving concretness for the general results.

Weaknesses

1- Lack of a significant breakthrough on a previously-identified and long-standing research discovery (i.e., dissipative freezing)

Report

I read carefully all the versions of the manuscripts and the previous referee's comments. I find the topic interesting and the results worthy of publication and I acknowledge the improvements made by the authors, but I do stick to the opinion of Referee 2 on 2022-5-19 that the manuscript will be more suited for SciPost Physics Core.

Indeed, I do not really see how the present manuscript meets at least one of the "Expectations" criteria of SciPost Physics:

1. There is no "groundbreaking theoretical/experimental/computational discovery", since dissipative freezing was already discovered in a previous work.

2. I do not find that the manuscript "presents a breakthrough on a previously-identified and long-standing research stumbling block" (i.e. dissipative freezing). Indeed, even if Ref [24] was only presenting dissipative freezing in a particular case, it was already clear that the presence of a strong symmetry was a key ingredient. In the present manuscript, the authors provide significant mathematical and physical insight into why freezing will emerge, but I would not qualify it as a breakthrough. It is rather an elaboration and a generalisation of a previously found result.

3. I do not find either that this manuscript "opens a new pathway in an existing or a new research direction, with clear potential for multipronged follow-up work". The manuscript provides a way to identify better whether or not a system undergoes dissipative freezing, but I would not qualify it as a new research direction: I do not find the manuscript impactful and helpful enough for the scientific community for guiding researchers in new research directions. How could one use the results of this paper for this purpose ?

4. I do not find either that it "provides a novel and synergetic link between different research areas". They identify a direct connection between rigorous mathematical results on ergodicity in open systems and dissipative freezing, but again there was already the idea in [24] thet dissipative freezing was an non-ergodic phenomenon. Also, the synergy between different fields is unclear.

Hence, while the present manuscript constitutes a significant elaboration on the mechanisms underlying dissipative freezing, I do not think it satisfies the publication criteria of SciPost Physics and how it would open significant research directions for the readership. However, I find that it clearly meets more adequately all the acceptance criteria of SciPost Physics Core:

1. "Address an important (set of) problem(s) in the field using appropriate methods with an above-the-norm degree of originality;"
2. "Detail one or more new research results significantly advancing current knowledge and understanding of the field."

and I would thus recommend it for publications in SciPost Physics Core, except if the authors can revise their manuscript so to provide materials that convince more strongly on how their manuscript meets the acceptance criteria of SciPost Physics.

Requested changes

- I suggest (as Referee 1 on 2022-8-9) a typo check.

-In the caption of Figure 2, it is said \gamma = 4 \omega. To make clearer the connection with Eq. (1), the number M of jump operators and/or the index j of \gamma should be specified.

- In the example 3.3, the authors work with qudits but below Eq. (29) mentions the "spin-3/2" operator. The dimension of the qudits used in the example should be mentioned more clearly.

  • validity: good
  • significance: good
  • originality: good
  • clarity: good
  • formatting: excellent
  • grammar: reasonable

Author:  Joseph Tindall  on 2022-09-23  [id 2844]

(in reply to Report 2 on 2022-09-04)

We are thankful for the thorough reading and comments on the manuscript. We have made the minor changes requested by the referee.

---

## Round 3 · Author Response

Dear Editor,

We have fixed the two minor errors pointed out by the referee.

On behalf of the authors,
Joseph Tindall

---

## Round 3 · List of Changes

1. Eq.(10) has now had the $m \in ...$ part removed.
2. A prime on the second $\beta$ indice has been added to $\rho_{R} ...$ which appears in Section 2.4.

---

## Round 4 · List of Changes

1) Fixed a typo concerning the α’s below Eq. (14) 2) Fixed a typo in Section 2.5 3) We have notated the freeze-time as $t_{f}$. 4) We have discussed that determining Eq. (26) generally requires diagonalization of the full Liouvillian. We have noted, however, that there exists a more simple case – based on simple relations involving the Lindblad and Hamiltonian operators – where it is known to be true. Moreover, we have reemphasized our argument that our results demonstrate that a single trajectory can demonstrate clear evidence of Eq. (26) – which is much more computationally efficient than diagonalization of the full Liouvillian. 5) We have removed the circular argument on page 17. 6) We have made it clear, in Section 2.4, that the steady state in a given symmetry subspace need not be unique. 7) We have made the argument (that one of $p^{(i)} (α_{1},t)$ and $p^{(i)} (α_{2},t)$ will be of order $\epsilon^{2}$ at the freeze-time) at the beginning of page 11 clearer. 8) We have made extensive grammatical changes in line with the referee’s suggestions. 9) We have removed the mention of qudits and made it clear we are referring to two particles of spin 3/2. We have also specified the Hilbert space dimension. 10) We have made the caption of Fig. 2 clearer and in-line with Eq. (1).

---

## Editorial Decision

published